# Immunoediting is not a primary transformation event in a murine model of MLL-ENL AML

Monika Dudenhöffer-Pfeifer[1], David Bryder[1,2]

Although it is firmly established that endogenous immunity can prevent cancer outgrowth, with a range of immunomodulatory strategies reaching clinical use, most studies on the topic have been restricted to solid cancers. This applies in particular to cancer initiation, where model constraints have precluded investigations of immunosurveillance and immunoediting during the multistep progression into acute myeloid leukemia (AML). Here, we used a mouse model where the chimeric transcription factor MLL-ENL can be conditionally activated in vivo as a leukemic "first-hit," which is followed by spontaneous transformation into AML. We observed similar disease kinetics regardless of whether AML developed in WT or immunocompromised hosts, despite more permissive preleukemic environments in the latter. When assessing transformed AML cells from either primary immunocompetent or immunocompromised hosts, AML cells from all sources could be targets of endogenous immunity. Our data argue against immunoediting in response to selective pressure from endogenous immunity as a universal primary transformation event in AML.

## Introduction

Acute myeloid leukemia (AML) is a highly aggressive form of blood cancer that emanates from hematopoietic progenitor cells arrested in differentiation. A precursor to AML arises at some point in time because of DNA mutations or other (epi)genetic events. Subsequent disease progression is the consequence of additional acquired molecular events of the founder clone, which selects for more aggressive subclones (Greaves & Maley, 2012). Recent sequencing studies have revealed that AML associates with fewer mutations than most other cancers (Kandoth et al, 2013; Lawrence et al, 2013), although identifiable driver mutations can almost always be identified (Ley et al, 2013).

It has been a long-standing idea that spontaneously arising cancer cells for the most part are eliminated by the endogenous immune system (Burnet, 1957), and CD8+ cytotoxic T cells and NK cells in particular. This can be easily envisioned with virus-driven tumors, in which immunity is directed to foreign viral antigens (Klein, 2009). However, immunity also develops against cancers with an endogenous origin. The key proposed mechanisms include the elimination of cancerous cells via neo- or other tumor-associated antigens that arise as a consequence of mutations and/or alternative molecular changes (DuPage et al, 2012; Matsushita et al, 2012) or the prevention of formation of tumor-promoting environments (Schreiber et al, 2011). In this view, tumor progression represents a continuous battle between endogenous immunity and developing preleukemic cells. Once precancerous cells have acquired properties that permit escape from such immunity, referred to as immunoediting, they can persist and acquire additional changes necessary to develop into overt cancer (Mittal et al, 2014). This concept integrates that development into cancer is rare, even in situations of excessive exposure to precancerous lesions (Klein, 2009), and that solid tumors arising in immunocompromised settings tend to be more immunogenic than those from immunocompetent settings (O'Sullivan et al, 2012). Although some aspects of immunoediting have been challenged (Willimsky & Blankenstein, 2005; Ciampricotti et al, 2012), escape from immunity is today regarded as one of the hallmarks of cancer (Hanahan & Weinberg, 2011).

Other restrictions of the immunoediting concept concern its generality. The overall low number of mutations in AML compared with other cancers might be particularly relevant (Kandoth et al, 2013; Lawrence et al, 2013) as it suggests that the formation of neoantigens is also more restricted. Despite evidence that AML can be susceptible to both adaptive and innate immune cell targeting (Austin et al, 2016), studies in patients exclusively characterize immune cell aspects in late-stage AMLs and/or in response to treatment. Thus, knowledge on the stepwise modulation of the immune responses that accompany AML progression is for the most part lacking. This includes whether immunoediting might be a primary mechanism of leukemia initiation. Although studying relapse could be argued to be very different, as it allows for comparison between two or more successive states of the disease, the initial comparator—the ground state at diagnosis—might still represent tumor clone/s that have evolved over time in response to

[1]Division of Molecular Hematology, Department of Laboratory Medicine, Lund University, Lund, Sweden [2]Sahlgrenska Cancer Centre, Gothenburg University, Gothenburg, Sweden

Correspondence: david.bryder@med.lu.se

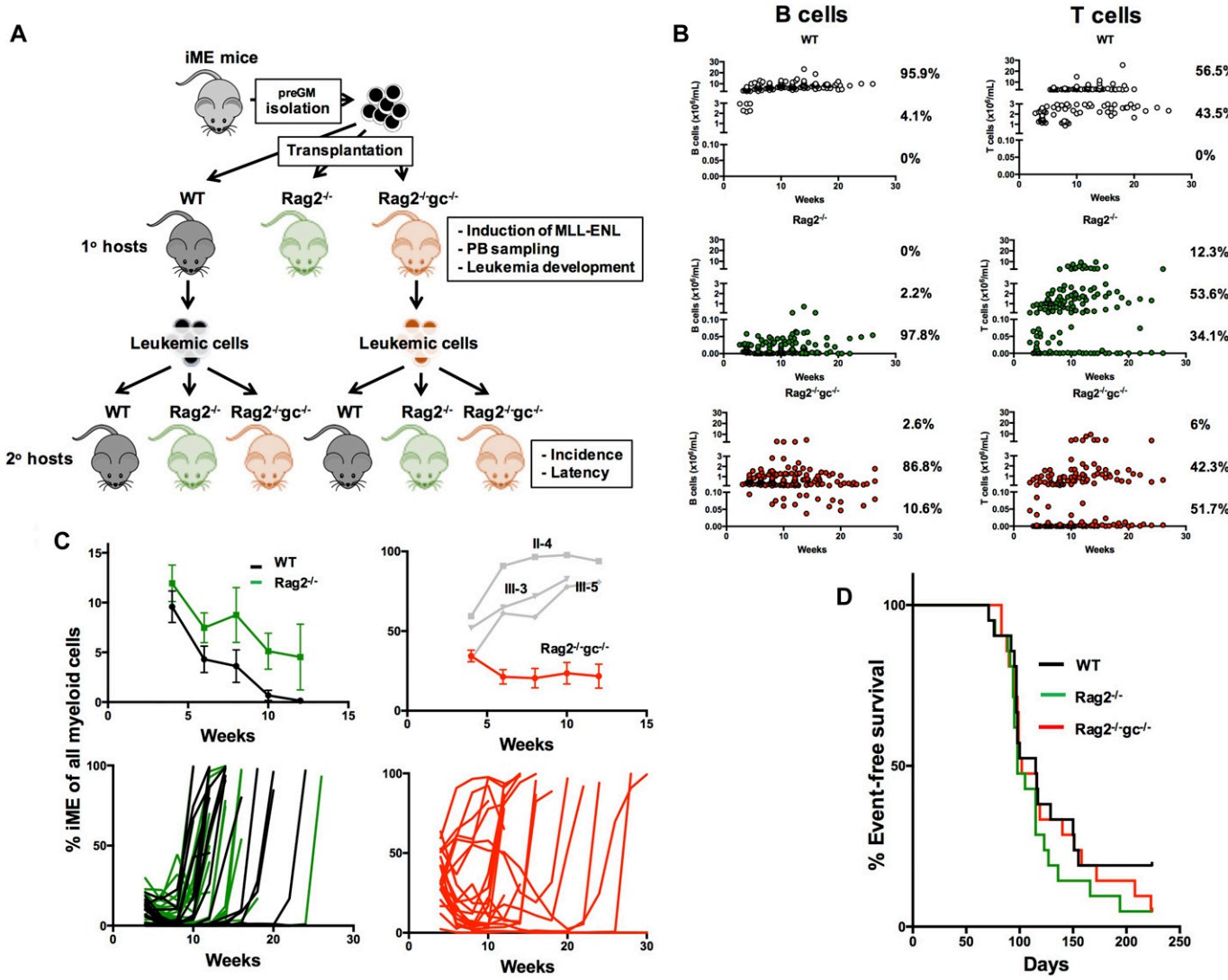

**Figure 1. The influence of immunity on transformation into MLL-ENL–driven AML.**
**(A)** Overview of the experimental design. 1,000 preGMs were isolated from iME mice and transplanted into sublethally (350 cGy) irradiated WT, Rag2$^{-/-}$, and Rag2$^{-/-}$gc$^{-/-}$ mice. Upon disease development, splenic AML cells were isolated from WT and Rag2$^{-/-}$gc$^{-/-}$ mice and transplanted at different doses (50,000, 100,000, or 300,000 leukemic cells) into sets of new sublethally irradiated WT, Rag2$^{-/-}$, and Rag2$^{-/-}$gc$^{-/-}$ hosts. **(B)** Total B or T cell numbers derived from iME preGMs in WT, Rag2$^{-/-}$, and Rag2$^{-/-}$gc$^{-/-}$ hosts during leukemia development. **(C)** The pattern of leukemia formation in WT (black lines) and Rag2$^{-/-}$ (green lines) hosts. Top graph, left: the amount of iME myeloid cells as a fraction of all myeloid cells in hosts, before leukemia development. Shown are the means ± SE. Bottom graph, left: the iME myeloid contribution in all WT and Rag2$^{-/-}$ hosts throughout the experiment (n = 21 recipients per group, from three separate experiments). Each line depicts an individual recipient. The pattern of leukemia formation in Rag2$^{-/-}$gc$^{-/-}$ hosts. Top graph, right: the amount of iME myeloid cells as a fraction of all myeloid cells in hosts, before leukemia development. Shown are the means ± SE. The values in three individual mice in which iME expanded excessively are indicated in grey (these mice were censored from the summary represented by the red curve). Bottom graph, right: the iME myeloid contribution in all Rag2$^{-/-}$gc$^{-/-}$ hosts throughout the experiment (n = 21 recipients, from three separate experiments). Each line depicts an individual recipient. **(D)** Kaplan–Meier survival curves of all transplanted primary mice. No significant difference in survival or disease latency was observed between any of the groups.

selective pressures such as immune evasion. A key experimental requirement to address primary transformation events is, therefore, the availability of models that allow for monitoring of the different phases of cancer development, starting from normal cells.

Chromosomal translocations that result in fusion proteins with aberrant transcriptional activities are often initiating events in AML (Estey & Döhner, 2006). These include fusions involving the *MLL1/KMT2A* gene. *MLL1* translocations comprise 35–50% of AML cases in infants. In older children and adults, they account for ~10% of all acute leukemias (Winters & Bernt, 2017). In general, patients with

MLL1 fusions have a poor prognosis and are treated according to high-risk protocols. Several mouse models have been used to model human MLL1 fusion–driven AML (Milne, 2017). We engineered a transgenic mouse model with a doxycycline-regulated human *MLL-ENL* fusion gene (iME mice) (Ugale et al, 2014). This enables an experimental strategy where defined hematopoietic progenitor cells can be isolated from WT mice—in our case, uninduced iME cells. A leukemic "first-hit" is mimicked by inducing MLL-ENL expression in iME cells transplanted into WT hosts, followed by monitoring of disease progression. Leveraging on our previous

observations using this model (Ugale et al, 2014)—that AML development in the iME model is characterized by an initial preleukemic expansion, a contraction phase, and not until thereafter transformation into aggressive AML—we here entertained that the iME model represents a relevant system to address the question of immunoediting in AML.

We show that immunoediting during transformation into MLL-ENL–driven AML is limited. Rather, the immune escape of arising disease associated with properties intrinsic to individual leukemic clones and could also be observed when AML developed in settings of reduced or absent adaptive and NK cell–mediated immunity. The immunogenicity to established AML was mediated by CD8[+] cells which rapidly developed signs of exhaustion during propagation of AML.

# Results and Discussion

### The influence of host immunity on transformation into MLL-ENL–driven AML

We designed an approach to study the influence of the immune system on the recognition, eradication, and potential immunoediting of (pre)leukemic cells during transformation into AML (Fig 1A). In this, we transplanted 1,000 BM granulocyte–monocyte progenitors (preGMs; Lin⁻Sca1⁻ckit⁺CD105⁻CD150⁻FcgR⁻) from iME (Ugale et al, 2014) mice into minimally conditioned (350 cGy) WT, Rag2$^{-/-}$ (Shinkai et al, 1992), and Rag2$^{-/-}$gc$^{-/-}$ (Cao et al, 1995) hosts (Fig 1A) that were continuously fed with doxycycline to induce MLL-ENL expression. PreGMs are potent AML-initiating cells in this model (Ugale et al, 2014). Rag2$^{-/-}$ mice lack B and T cells, whereas Rag2$^{-/-}$gc$^{-/-}$ mice in addition lack NK cells. This combination of recipients was chosen to broadly assess the contribution of adaptive and innate immunity, with NK cells previously shown to prominently influence immunoediting in solid tumors (O'Sullivan et al, 2012) and established AML (Lion et al, 2012). The presence of iME-derived myeloid blood cells was thereafter evaluated every second week, along with disease parameters indicative of transformation. Upon disease in primary hosts, we extracted leukemic cells from WT and Rag2$^{-/-}$gc$^{-/-}$ hosts and transplanted these into secondary recipients, again represented by a combination of immunocompetent and immunodeficient strains (Fig 1A).

Although predominantly myeloid restricted, the population of preGMs harbor also some lymphoid potential in vitro (Pronk et al, 2008). To evaluate the differentiation ability of preGMs in vivo, we analyzed B and T lymphocytes in the transplanted immunodeficient and WT hosts. The lymphoid contribution in each mouse and at each analysis time point was stratified into three different levels, with low cell numbers defined as up to 0.1 × 10$^6$ cells/ml, intermediate levels between 0.5 and 3 × 10$^6$ cells/ml, and high levels between 5 and 30 × 10$^6$ cells/ml. B lymphocytes were generated in both types of immunodeficient hosts, with a substantially higher contribution in Rag2$^{-/-}$gc$^{-/-}$ mice. However, in contrast to the WT hosts, the B cell concentrations were much lower in both types of immunodeficient mice (WT: 95.9% of mice with high levels and 4.1% of mice with intermediate levels; Rag2$^{-/-}$: 0% of mice with high

levels, 2.2% of mice with intermediate levels, and 97.8% with low levels; and Rag2$^{-/-}$gc$^{-/-}$: 2.6% with high levels, 86.8% with intermediate levels, and 10.6% of mice with low levels). For T lymphocytes, the levels in both immunodeficient hosts were similar but far below the magnitudes observed in WT hosts (WT: 56.5% with high levels, 43.5% with intermediate levels, and 0% with low levels; Rag2$^{-/-}$: 12.3% with high levels, 53.6% with intermediate levels, and 34.1% with low levels; and Rag2$^{-/-}$gc$^{-/-}$: 6% with high levels, 42.3% with intermediate levels, and 51.7% with low levels) (Fig 1B).

In agreement with our previous work (Ugale et al, 2014) and despite the very different conditioning regimen used here (low-dose versus high-dose/lethal irradiation), we could in both WT and Rag2$^{-/-}$ primary hosts observe three phases of AML development. We first observed an initial expansion of iME-derived cells (WT/uninduced preGMs do not produce detectable myeloid offspring 4 wk after transplantation [Ugale et al, 2014]). This was followed by a contraction and thereafter leukemic development (Fig 1B and D). The mortality from AML was asynchronous in between mice and typically developed after 8 wk and onward (Fig 1D) and could for the most part be predicted by prior blood sampling results (Fig 1C). Although the magnitude of donor myeloid cells was initially similar in WT and Rag2$^{-/-}$ hosts, we observed the trend that the contraction preceding transformation was stronger in WT mice (Fig 1C).

The situation was strikingly different in Rag2$^{-/-}$gc$^{-/-}$ hosts. These presented with substantially higher contribution of iME myeloid cells at the first 4-wk analysis time point (34.2 ± 3.7% compared with 9.6 ± 1.6% and 11.9 ± 1.8% in WT and Rag2$^{-/-}$ mice, respectively), which for the most part remained high until development of AML (Fig 1C–D). Of the 20 Rag2$^{-/-}$gc$^{-/-}$ mice in which leukemia developed, 3 displayed very high initial numbers of donor myeloid cells at the first analysis point (4 wk), which continued to elevate until the mice became moribund (Fig 1C). No such instance was observed in WT or Rag2$^{-/-}$ hosts. However, despite these pronounced differences, the disease latency was highly similar between the three evaluated groups (median survival 115, 98, and 102 d for WT, Rag2$^{-/-}$, and Rag2$^{-/-}$gc$^{-/-}$, respectively; Fig 1B). Disease incidence was also not significantly different among the groups, although leukemia failed to develop in a few more WT recipients (WT 4/21, Rag2$^{-/-}$ 1/21, and Rag2$^{-/-}$gc$^{-/-}$ 1/21 evaluated mice).

### Approaching immunoediting as a primary mechanism for transformation into AML

If immunoediting would be applicable to the setting of AML evaluated here, established leukemia from primary immunodeficient hosts should be more immunogenic than those from WT hosts (O'Sullivan et al, 2012). Therefore, AML developing in immunodeficient primary hosts should either propagate slower or fail to propagate at all when evaluated in secondary WT hosts, but effectively propagate in immunodeficient hosts. To evaluate this, we next isolated varying doses of leukemic cells (50,000, 100,000, or 300,000 cells) obtained from six WT and six Rag2$^{-/-}$gc$^{-/-}$ primary hosts (Fig 2A). The cells were next transplanted into 350-cGy irradiated WT, Rag2$^{-/-}$, or Rag2$^{-/-}$gc$^{-/-}$ hosts (Figs 1A and S1 and Table 1). As transplantation of established leukemia, as opposed to the time of primary transformation, leads to more rapid disease development (Ugale et al, 2014), we monitored survival in these hosts up to 70 d.

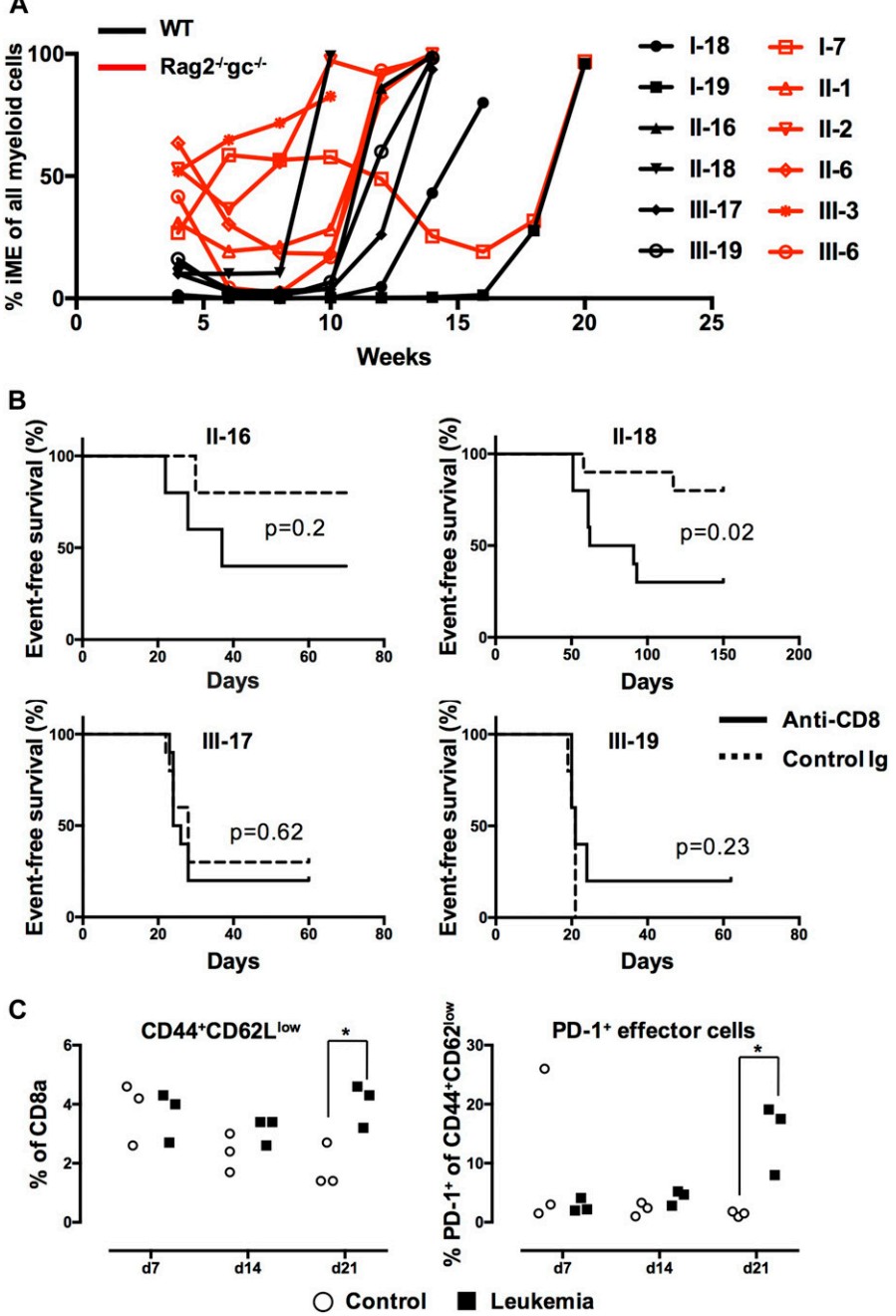

**Figure 2. CD8+ T cells underlie the rejection of primary AML in secondary hosts.**
**(A)** The iME contribution to myeloid reconstitution in primary hosts in the 12 separate cases of AML evaluated in secondary recipients (Table 1). Lines represent individual primary mice from which leukemic cells were extracted (black = WT host and red = Rag2$^{-/-}$gc$^{-/-}$ hosts). **(B)** Kaplan–Meier curves depicting AML propagation as a consequence of CD8 depletion. Four different leukemic clones were evaluated by transplantation of 50,000 (III-17 and III-19) or 100,000 (II-16 and II-18) AML cells into WT mice. Graphs depict 5 mice per group for clones II-16 and III-19, and 10 mice in each group for clones II-18 and III-17. **(C)** Sublethally (350-cGy) irradiated WT mice were transplanted with 300,000 AML cells from clone II-16. After 7, 14, and 21 d, the mice were euthanized and the frequency of splenic CD8+ effector cells (CD44$^{hi}$CD62L$^{low}$) analyzed. CD8+ effector cells were in addition analyzed for PD-1 (as an indicator of exhaustion). Data points represent values from individual mice (n = 3 per time point). *P < 0.05.

Clone I-7, which had an origin in Rag2$^{-/-}$gc$^{-/-}$ hosts, failed to be propagated in either type of secondary host over the reported time period (Table 1). We observed a substantial variation in disease incidence and latency between the remaining 11 evaluated clones (Table 1). This was in general also dependent on the amount of transplanted leukemic cells, with more cells transplanted leading to shorter latency. A more striking observation was that the disease more often failed to arise in secondary WT hosts, whereas immunodeficient recipients receiving the same cells and dose succumbed to the disease (Table 1 and Fig S1). The responses in both types of immunodeficient mice studied were similar, demonstrating

that the effect was not restricted to NK cells. Importantly, however, these findings reflected the immunostatus of the secondary rather than the primary hosts in which the leukemia originally developed (Table 1).

Given the efficient propagation of AML in both secondary Rag2$^{-/-}$ and Rag2$^{-/-}$gc$^{-/-}$ mice (Table 1 and Fig S1), we suspected that the rejection in WT hosts involved CD8+ cytotoxic T cells, given their prominent role in tumor biology (Gajewski et al, 2013; Teng et al, 2015). To test this, we next used an antibody-based strategy to deplete WT mice of CD8+ cells (Weiss & Jiang, 2012) before transplantation of AML cells. As responses to established leukemic cells

**Table 1. Leukemia latency and penetrance in secondary hosts.**

| 1° Hosts | Clone | 2° Hosts | Transplanted cell number | Dead | Survival (d) | 1° Hosts | Clone | 2° Hosts | Transplanted cell number | Dead | Survival (d) |
|---|---|---|---|---|---|---|---|---|---|---|---|
| WT | I-18 | $Rag2^{-/-}$ $gc^{-/-}$ | 50,000 | **1/3** | 64, >70, >70 | $Rag2^{-/-}$ $gc^{-/-}$ | I-7 | $Rag2^{-/-}$ $gc^{-/-}$ | 50,000 | **0/3** | >70, >70, >70 |
| | | | 100,000 | **1/3** | 68, >70, >70 | | | | 100,000 | **0/3** | >70, >70, >70 |
| | | $Rag2^{-/-}$ | 50,000 | **2/3** | 48, 68, >70 | | | $Rag2^{-/-}$ | 50,000 | **0/3** | >70, >70, >70 |
| | | | 100,000 | **0/3** | >70, >70, >70 | | | | 100,000 | **0/3** | >70, >70, >70 |
| | | WT | 300,000 | **0/3** | >70, >70, >70 | | | WT | 300,000 | **0/3** | >70, >70, >70 |
| | I-19 | $Rag2^{-/-}$ $gc^{-/-}$ | 50,000 | 3/3 | 35, 46, 49 | | II-1 | $Rag2^{-/-}$ $gc^{-/-}$ | 50,000 | 3/3 | 26, 27, 28 |
| | | | 100,000 | 3/3 | 32, 33, 49 | | | | 100,000 | 3/3 | 26, 28, 28 |
| | | $Rag2^{-/-}$ | 50,000 | 4/4 | 38, 53, 63, 68 | | | $Rag2^{-/-}$ | 50,000 | 4/4 | 29, 29, 30, 31 |
| | | | 100,000 | 3/3 | 38, 38, 38 | | | | 100,000 | 3/3 | 26, 28, 28 |
| | | WT | 50,000 | **0/5** | >70, >70, >70, >70, >70 | | | WT | 50,000 | 2/3 | 51, 67, >70 |
| | | | 100,000 | **0/5** | >70, >70, >70, >70, >70 | | | | 100,000 | 2/3 | 42, 46, >70 |
| | | | 300,000 | 2/5 | 32, 34, >70, >70, >70 | | | | 300,000 | 3/3 | 25, 26, 39 |
| | II-16 | $Rag2^{-/-}$ $gc^{-/-}$ | 50,000 | 3/3 | 22, 31, 31 | | II-2 | $Rag2^{-/-}$ $gc^{-/-}$ | 50,000 | 4/4 | 33, 33, 34, 34 |
| | | | 100,000 | 4/4 | 23, 24, 24, 30 | | | | 100,000 | 4/4 | 28, 30, 31, 32 |
| | | $Rag2^{-/-}$ | 50,000 | 4/4 | 29, 35, 35, 35 | | | $Rag2^{-/-}$ | 50,000 | 3/3 | 28, 29, 29 |
| | | | 100,000 | 3/3 | 30, 32, 32 | | | | 100,000 | 4/4 | 28, 31, 31, 33 |
| | | WT | 50,000 | **0/3** | >70, >70, >70 | | | WT | 50,000 | 3/3 | 29, 31, 41 |
| | | | 100,000 | **2/4** | 30, 38, >70, >70 | | | | 100,000 | 4/4 | 28, 29, 32, 35 |
| | | | 300,000 | **3/4** | 27, 27, 48, >70 | | | | 300,000 | **4/5** | 28, 30, 30, 35, >70 |
| | II-18 | $Rag2^{-/-}$ $gc^{-/-}$ | 50,000 | 3/3 | 43, 52, 55 | | II-6 | $Rag2^{-/-}$ $gc^{-/-}$ | 50,000 | 3/3 | 36, 37, 37 |
| | | | 100,000 | 3/3 | 43, 54, 55 | | | | 100,000 | 3/3 | 30, 34, 36 |
| | | $Rag2^{-/-}$ | 50,000 | 3/3 | 44, 64, 64 | | | $Rag2^{-/-}$ | 50,000 | 4/4 | 38, 43, 43, 43 |
| | | | 100,000 | 3/3 | 51, 57, 59 | | | | 100,000 | 3/3 | 42, 45, 46 |
| | | WT | 50,000 | **0/3** | >70, >70, >70 | | | WT | 50,000 | **0/2** | >70, >70 |
| | | | 100,000 | **1/3** | 63, >70, >70 | | | | 100,000 | **0/3** | >70, >70, >70 |
| | | | 300,000 | 2/3 | 45, 57, >70 | | | | 300,000 | 2/2 | 35, 35 |
| | III-17 | $Rag2^{-/-}$ $gc^{-/-}$ | 50,000 | 3/3 | 25, 25, 25 | | III-3 | $Rag2^{-/-}$ $gc^{-/-}$ | 50,000 | 4/4 | 25, 25, 25, 25 |
| | | | 100,000 | 3/3 | 21, 22, 22 | | | | 100,000 | 3/3 | 21, 25, 28 |
| | | $Rag2^{-/-}$ | 50,000 | 3/3 | 24, 25, 25 | | | $Rag2^{-/-}$ | 50,000 | 3/3 | 25, 25, 25 |
| | | | 100,000 | 3/3 | 25, 25, 26 | | | | 100,000 | 3/3 | 24, 25, 28 |
| | | WT | 50,000 | **2/3** | 24, 28, >70 | | | WT | 50,000 | 3/3 | 27, 27, 46 |
| | | | 100,000 | 3/3 | 28, 35, 36 | | | | 100,000 | **2/3** | 27, 28, >70 |
| | | | 300,000 | **4/6** | 19, 30, 30, 40, >70, >70 | | | | 300,000 | 6/6 | 23, 26, 27, 31, 32, 34 |
| | III-19 | $Rag2^{-/-}$ $gc^{-/-}$ | 50,000 | 4/4 | 20, 22, 22, 23 | | III-6 | $Rag2^{-/-}$ $gc^{-/-}$ | 50,000 | 3/3 | 44, 52, 60 |
| | | | 100,000 | 4/4 | 22, 22, 22, 22 | | | | 100,000 | 3/3 | 32, 32, 46 |
| | | $Rag2^{-/-}$ | 50,000 | 3/3 | 23, 23, 23 | | | $Rag2^{-/-}$ | 50,000 | 4/4 | 37, 51, 63, 65 |

**Table 1. Continued**

| | | | | | | | | |
|---|---|---|---|---|---|---|---|---|
| | | 100,000 | 4/4 | 20, 20, 24, 24 | | 100,000 | 4/4 | 33, 38, 48, 48 |
| | WT | 50,000 | 3/3 | 20, 22, 25 | WT | 50,000 | **2/3** | 54, 65, >70 |
| | | 100,000 | 3/3 | 20, 20, 21 | | 100,000 | 4/4 | 48, 50, 50, 51 |
| | | 300,000 | 5/5 | 20, 20, 21, 23, 25 | | 300,000 | **4/6** | 36, 44, 57, 59, >70, >70 |

50,000, 100,000, or 300,000 primary leukemic cells from WT or Rag2$^{-/-}$gc$^{-/-}$ hosts (Fig 2A) were transplanted to secondary WT or immunodeficient environments and monitored over 70 d. Depicted is the origin of leukemia, type of secondary host, the assessed cell numbers, the incidence of mortality, and the time until death. Bold entries indicate groups with surviving mice.

were highly clone dependent (Table 1), we evaluated four independent clones with a WT host origin (Fig 2A). These clones varied in both disease latency and potential to propagate AML in secondary WT hosts but displayed an absolute disease penetrance in secondary immunodeficient hosts (Table 1). For two of the evaluated clones, II-16 and II-18, we observed that CD8 depletion accelerated both disease progression and incidence (Fig 2B). By contrast, CD8 depletion had little or no effect on the disease formation from clones III-17 and III-19 (Fig 2B).

We finally investigated CD8$^+$ T cell subsets during propagation of leukemia in secondary hosts. For this, we focused on the responses to clone II-16, a clone which could be eliminated in a CD8$^+$-dependent manner (Fig 2A and B, and Table 1). Mice that received cells from the same clone, but following withdrawal of MLL-ENL expression, served as controls. The progression of leukemia at the different time points was evaluated by analyzing the amount of donor myeloid cells in the spleen (Fig S2). Although the splenic cellularity and distributions of CD8 cells were only marginally different 1 and 2 wk after transplantation (data not shown), we observed a dramatic increase in overall splenic cellularity at week 3 in the MLL-ENL–induced group. This reflected mainly an expansion of leukemic cells (Fig S2) but coincided also with an increase in CD8$^+$ effector cells (Fig 2C), of which many expressed PD-1, an indicator of CD8$^+$ T cell exhaustion (Pauken et al, 2016).

Although hypotheses on immunosurveillance and immunoediting have been extensively investigated in solid cancer models (Teng et al, 2015) and evasion from immunity today is classified as a hallmark of cancer (Hanahan & Weinberg, 2011), there is limited information on these processes in AML. Here, we applied a mouse model in which an MLL-ENL fusion protein can be induced at the physiological expression level (Ugale et al, 2014). This allowed us to investigate the influence of immunity on AML initiation. By comparing an immunologically intact versus two different immunodeficient environments, we observed dramatically different patterns of disease formation. Preleukemic propagation in WT hosts was characterized by a strong contraction phase that preceded overt AML transformation. Although weaker, this contraction phase could also be observed in Rag2$^{-/-}$ hosts, whereas it was severely diminished in the stronger immunocompromised setting of Rag2$^{-/-}$gc$^{-/-}$ hosts. As a result, Rag2$^{-/-}$gc$^{-/-}$ mice presented with a substantially higher burden of candidate preleukemic donor cells early after transplantation. However, and somewhat surprising to us, the latency until overt transformation/disease was not different when compared with either WT or Rag2$^{-/-}$ hosts. Furthermore, when the immunogenicity of primary/established leukemias was evaluated in secondary hosts, the investigated AML clones behaved remarkably similar. This argues

against immunoediting as an evasion mechanism to endogenous immunity as a primary transformation mechanism. Rather, the changes leading to immune escape/overt transformation might either affect proliferation/apoptosis in an intrinsic manner, or rely on an interplay between transformation and appropriate microenvironments, where the latter might be rate limiting and independent of the immune mechanisms approached here.

Although not extensively investigated, a previous study approached the immunogenicity/editing to MLL-ENL AML using an alternative model of MLL-ENL–driven AML (Nakata et al, 2014). From that work, it was concluded that both adaptive immunity, but above all NK cells, could contribute to AML rejection (Nakata et al, 2014). We believe that the differences between that and our study make direct comparisons difficult. Whereas we aimed our work on immunoediting during the primary transformation process in vivo, Nakata et al (2014) studied cells that had been subjected to an in vitro transformation procedure. Other differences include retrovirus-mediated introduction of MLL-ENL, as compared with the conditional MLL-ENL allele in our transgenic model. We previously established that retroviral introduction of MLL-ENL leads to excessive doses of the fusion protein (Ugale et al, 2014), which is relevant for the transformation kinetics. It could also be anticipated that the chimeric transcription factor dosage represented by MLL-ENL could influence the levels and types of antigens presented, although immunogenicity to MLL-ENL as a neoantigen perhaps can be excluded (Nakata et al, 2014). Finally, whereas Nakata et al (2014) transplanted transformed cells into completely unconditioned (Rag2$^{-/-}$gc$^{-/-}$) hosts, the development of both iME leukemia from preGMs and the propagation of established leukemia in WT hosts required in our hands mild conditioning (data not shown). Although it would evidently be more optimal to avoid conditioning, we believe our data nonetheless demonstrate that the regimen used herein (350 cGy irradiation) preserves immune function at sufficient levels to approach the questions at hand.

Why could it then be that immunoediting appears to only marginally affect transformation into AML, while being dominant in models of solid cancer (DuPage et al, 2012; Matsushita et al, 2012)? One apparent aspect concerns the disseminated nature of AML. This should limit mechanisms that relate to physical properties of the tumor microenvironment, including changes that affect metabolism and/or the influence of immunosuppressive cells (Turley et al, 2015). The ability of such microenvironments to promote more extensive mutagenesis (Reynolds et al, 1996) could in turn lead to the formation of tumor antigens and might be linked to the low mutation frequency in AML as compared with other cancers (Kandoth et al, 2013; Lawrence et al, 2013). MLL-fusion leukemia is particularly noteworthy,

with childhood *MLL-AF4* translocations having the least number of secondary mutations of any leukemia (Andersson et al, 2015), although this is perhaps less obvious in adult *MLL*-rearranged AML (Grossmann et al, 2013).

In our work/model system, a main rejection mechanism was mediated by CD8$^+$ T cells, which is in line with their strong anti-tumor effect across cancer types (Gajewski et al, 2013; Teng et al, 2015). Multiple mechanisms exist for the inactivation of antitumor activity of CD8$^+$ cells, including recruitment of suppressor cells, poor co-stimulatory activation, and the down-regulation of histocompatibility antigens on tumor cells (Gajewski et al, 2013). Although the latter does not appear general in AML (Wetzler et al, 2001), it has been reported as a mechanism for relapse (Vago et al, 2009). We observed that CD8$^+$ effector T cells during AML propagation became gradually activated and rapidly evolved into cells with a PD-1$^+$ exhaustion phenotype as the leukemic burden became excessive, which seems to be in agreement with previous mechanisms of immune rejection in AML (Zhou et al, 2011).

It is well established that immunogenicity to AML can be harnessed for therapeutic benefit and that escape from such immunity can underlie disease relapse (Austin et al, 2016). Our results are not at odds with such findings. That not all primary leukemias were rejected in immunocompetent hosts in our work could very well reflect that host immunity was unable to catch up with the excessive AML propagation associated with some individual clones and/or in individual hosts. It has previously been demonstrated that MLL-AF9–bearing AML cells can escape immune rejection in a dose-dependent manner, despite expression of a strong tumor-associated antigen (Hasegawa et al, 2015). This points to a significant variation in leukemia-initiating cell frequency and/or behavior among individually arising AML clones, and is likely relevant for the interpretations of many murine MLL-fusion AML models (Milne, 2017) in which AML cells are often heavily propagated/selected before being studied.

## Materials and Methods

### Mice and in vivo MLL-ENL induction

Rag2$^{-/-}$ (Jax stock 008449) and gc$^{-/-}$ (Jax stock 003174) mice were acquired from Jackson Laboratories. These strains were crossed to generate the Rag2$^{-/-}$gc$^{-/-}$ strain. C57Bl/6J mice were purchased from Janvier Labs. All animals were bred and maintained at the animal facility at the Biomedical Medical Center at Lund University in accordance with local ethical regulations (ethical permit: M186-15). MLL-ENL induction in vivo was performed by administering doxycycline food (2 g/kg; ssniff Spezialdiät) 5 d before transplantation and throughout the experiments.

### Cell isolation and transplantation

For isolation of preGMs (Lin$^-$cKit$^+$CD41$^-$CD105$^-$CD150$^-$FcgRII/III$^-$), single-cell suspensions of BM cells from iME mice were lineage-depleted using biotinylated antibodies: B220 (RA3-6B2; BioLegend), CD4 (GK1.5; BioLegend), CD8a (53-6.7; BioLegend), CD11b (M1/70;

BioLegend), Gr-1 (RB6-8C5; BioLegend), and TER-119 (TER-119; BioLegend). Subsequently, cells were isolated by magnet sorting according to the manufacturer's instructions (Miltenyi Biotech). The negatively isolated cells were stained with CD117 APC/Alexa780 (2B8; BioLegend), CD150 APC (TC15-12F12.2; BioLegend), CD105 PE/Cy7 (MJ7/18; BioLegend), CD16/32 Alexa700 (93; eBioscience), Sca-1 Pacific Blue (E13-161.7; BioLegend), and fluorescence-labeled streptavidin to exclude remaining lineage cells. The cells were sorted on a FACS Aria II or III at the shared FACS facility at the Lund Stem Cell Center.

For transplantation experiments using preGMs, 1,000 iME cells were transplanted into 350-cGy irradiated 10- to 12-wk-old female recipient mice. Single-cell suspensions from spleens of leukemic WT and Rag2$^{-/-}$gc$^{-/-}$ mice were frozen in freezing media (70% FCS, 10% DMSO, and 20% DMEM) and stored at −150°C until used for further experiments. A log-rank Mantel–Cox test was used to assess significance in survival and latency (GraphPad Prism, SD).

For transplantations of transformed AML cells, splenocytes were thawed and host, B, T, and NK cells depleted using biotinylated anti-mouse CD45.2 (104; BioLegend), anti-mouse CD19 (1D3; eBioscience), anti-mouse CD4 (GK1.5; BioLegend), anti-mouse CD8 (53-6.7; BioLegend), and anti-mouse NK1.1 (PK136; BioLegend). The indicated leukemic cell numbers were transplanted into 350-cGy irradiated 10- to 12-wk-old female recipient mice.

Mice were recorded as leukemic if they had >90% donor myeloid cells or displayed other signs of sickness, including inactivity, anemia, and/or motor dysfunction. When possible, leukemia was verified by necroscopic inspection, with pronounced splenomegaly as a direct disease correlator. When WT or Rag2$^{-/-}$ mice displayed a >twofold increase in myeloid donor cells between two adjacent time points (2-wk interval), they in almost all cases developed fatal leukemia 2–6 wk later. This was a more difficult predictor in Rag2$^{-/-}$gc$^{-/-}$ mice (Fig 1C).

### In vivo CD8$^+$ cell depletion

To deplete CD8$^+$ T cells in vivo, mice were injected intraperitoneally with 100 µg rat anti-mouse CD8$^+$ antibodies (53-6.72; BioXCell) at days −10, −8, 21, and 28, where day 0 represents the day of transplantation. Control groups were administered 100 µg rat IgG2a isotype control (2A3; BioXCell) on the same days. The efficiency of CD8$^+$ cell depletion was verified on day −1 on peripheral blood cells, by staining with FITC rat anti-mouse CD8b.2 antibodies (53-5.8; Sony) and subsequent FACS analysis (data not shown).

### Peripheral blood preparation

Peripheral blood was collected from the tail vein in Microvette tubes (Sarstedt) and cellularity was analyzed using Sysmex XE-5000. Erythrocytes were sedimented with 1% dextran T500 (Sigma-Aldrich) and remaining erythrocytes were lysed with ACK (0.15 M NH$_4$Cl, 10 mM KHCO$_3$, and 0.1 mM EDTA; pH 7.2–7.4). The cells were stained and analyzed as described (Ugale et al, 2017). The absolute lymphocyte counts were determined by multiplying the total white blood cell counts and the frequencies of B or T cells established by the FACS analysis.

## CD8 subset analysis

CD8$^+$ T cell subsets were stained with CD11c BV570 (N418; BioLegend) (negative marker), CD8a PerCP/Cy5.5 (53-6.7; BioLegend), CD44 PE (IM7; BD Biosciences), CD62L Alexa488 (MEL-14; Sony), CD127a BV510 (A7R34; Sony), PD-1 BV786 (29F.1A12; BioLegend), and CD69 PE/Cy7 (H1.2F3; BioLegend) and analyzed on an LSR Fortessa or an LSRII (Becton Dickinson). Significance of the frequency of effector CD8$^+$ cells and the % PD-1$^+$ CD8$^+$ effector cells was assessed using a paired *t* test (Excel; Microsoft).

# Supplementary Information

# Acknowledgements

This work was generously supported by a European Research Council (ERC) consolidator grant LEUKEMIABARRIER (615068), the Swedish Medical Research Council, the Swedish Cancer Society, the Tobias Foundation, and the Knut and Alice Wallenberg Foundation. We gratefully acknowledge Cornelis-Jan Pronk for critical reading of the manuscript and the expert technical assistance from Gerd Sten.

## Author Contributions

D Bryder: conceptualization, data curation, supervision, funding acquisition, investigation, and writing—original draft, project administration, and writing—review and editing.
M Dudenhöffer-Pfeifer: conceptualization, data curation, formal analysis, investigation, methodology, and writing—original draft, review, and editing.

## Conflict of Interest Statement

The authors declare that they have no conflict of interest.

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
