## [Reviewer comments · Life Science Alliance]

Immunoediting is Not a Primary Transformation Event in a Murine Model of MLL-ENL AML

Monika Dudenhöffer-Pfeifer and David Bryder

DOI: 10.26508/lsa.201800079

Review timeline:

Submission Date:	25 April 2018
Editorial Decision:	9 May 2018
Revision Received:	11 June 2018
Editorial Decision:	26 June 2018
Accepted:	4 July 2018

Report:

(Note: Letters and reports are not edited. The original formatting of letters and referee reports may not be reflected in this compilation.)

1st Editorial Decision

9 May 2018

Thank you for submitting your manuscript entitled "Immunoediting is Not a Primary Transformation Event in a Murine Model of MLL-ENL AML" to Life Science Alliance. The manuscript was assessed by expert reviewers, whose comments are appended to this letter. We invite you to submit a revision if you can address the reviewers' key concerns, as outlined here.

As you will see, the reviewers appreciate your data, and reviewer #2 provides constructive input on how to further strengthen your data. During our reviewer cross-consultation session, reviewer #1 also mentioned that the issues raised by reviewer #2 should get addressed.

We think that the issues raised are feasible to address in a short time. Please note that secondary transplants in non-irradiated recipients are not required for acceptance here. Please get in touch in case you would like to discuss any of the revision points further.

Thank you for this interesting contribution to Life Science Alliance. We are looking forward to receiving your revised manuscript.

REFeree REPORTS

Reviewer #1 (Comments to the Authors (Required)):

The manuscript is based on a very interesting, timely and relevant research question, if immunoediting plays a role in cancer initiation of AML. I agree with the authors that the theory of immuno-editing is based on data in solid tumors and that the manuscript fills a knowledge gap. I appreciate that the authors specify in the title of the manuscript that it is about primary transformation, murine AML and MLL-ENL AML. Because in this way, the statement in the title is covered by the data in the manuscript, avoiding general AML statements that are not covered by the data.

In my opinion, no additional issues should be addressed.

Reviewer #2 (Comments to the Authors (Required)):

This is an interesting manuscript using a novel MLL-ENL inducible knockin mouse model. Immune responses to leukemia are seen in some mice, but the clonal differences between AML appears to

influence immune responsiveness. Of interest, MLL^{ENL} is usually found in ALL, but this model develops AML. Other MLL fusions are more commonly associated with AML.

Comments/ questions:

Are there differences in cooperative lesions found in immune responsive vs. non responsive clones (Table 1)?

All mice receive conditioning with irradiation, which is immunosuppressive. How can these experiments be interpreted in the context of WT recipients that are irradiated. Secondary transplants should be done in non irradiated recipients.

Do iME donor cells differentiate into mature T cells or B cells in Rag or Rag-Gc recipients?

Figure 2B: no statistics are provided, 2/4 donors showed acceleration with CD8 depletion

Withdrawal of MLL^{ENL} as a control - data are not provided to demonstrate disease regression. More information is needed.

Table 1 is difficult to interpret. KM survival curves for similar doses of cells should be provided, and perhaps a summary of the effects on survival. Infinite survival is not an appropriate way to represent survival, "not reached" would be more appropriate.

Immune-editing and immunoediting are used throughout. Consistency is recommended.

Fig 2C: was this experiment done multiple times, and is this representative data, or pooled results, or a single experiment only?

1st Revision – authors' response

11 June 2018

Query: Are there differences in cooperative lesions found in immune responsive vs. non responsive clones (Table 1)?

While an interesting question, we feel it is beyond the scope of the current manuscript. Little is known about the identity of cooperating mutations in murine AML. Therefore, to address this issue one would have to conduct whole genome sequencing (or alternatively exome sequencing) of a large number of samples (responsive vs non-responsive). In a separate ongoing project, we have attempted to identify secondary lesions in an immune- intact setting using either WGS or WES – this work has revealed several technical considerations, including the necessity to preserve matched control samples for each leukemic samples. Thus, even if we would have the resources at hand for such work (which is very expensive), it would require us to redo all the work from scratch (primary transplant, secondary transplants).

Query: All mice receive conditioning with irradiation, which is immunosuppressive. How can these experiments be interpreted in the context of WT recipients that are irradiated. Secondary transplants should be done in non-irradiated recipients.

We agree that it would have been more optimal to conduct experiments in a non- conditioned setting (an issue we also discuss in our manuscript). However, while we have some preliminary data that established leukemia can establish in non-irradiated Rag2^{-/-}gc^{-/-} mice (Dudenhöffer-Pfeifer, unpublished data 2017), we have not established that this would work in WT hosts. It is not evident that the different types of hosts can be compared, as considerations on physical space/engraftment capabilities come into play. As we observe pronounced differences in leukemic propagation (of same clones) in secondary hosts (differing in immune status), we argue the strategy we took preserves enough immunity to reveal differences/provide answers to the question at hand.

Query: Do iME donor cells differentiate into mature T cells or B cells in Rag or Rag-Gc recipients?

As a genuine lymphoid potential of the evaluated target cells would risk establishing a lymphoid-competent environment, this is a highly relevant question. To clarify this issue, we have in our revised manuscript added a new Figure 1B and accompanying text in the results section that provides information on the peripheral blood adaptive immune components in all primary mice at all evaluated time points. The data demonstrate that some amount of B and T lymphopoiesis can arise from the transplanted preGMs in immunocompromised hosts, although the magnitudes are considerably lower than those of WT hosts (and in many instances beyond detection).

Query: Figure 2B: no statistics are provided, 2/4 donors showed acceleration with CD8 depletion

We have in our revised manuscript added exact p-values to these graphs.

Query: Withdrawal of MLLENL as a control - data are not provided to demonstrate disease regression. More information is needed.

We previously demonstrated that leukemias in the iMLL-ENL model are strictly dependent on continuous MLL-ENL expression (Ugale et al, Cell reports 2014). This is quite well established for MLL-fusions (Horton et al, Blood May 14;113(20):4922-9.2009). To clarify further, we have in our revised manuscript added a new Supplementary Figure 2 that describes the expansion of the donor myeloid/leukemic cells in the presence or absence of MLL-ENL induction.

Query: Table 1 is difficult to interpret. KM survival curves for similar doses of cells should be provided, and perhaps a summary of the effects on survival. Infinite survival is not an appropriate way to represent survival, "not reached" would be more appropriate.

We agree that Table 1 contains very much data and is difficult to oversee. We spent a lot of time on discussing and looking at different ways for how to present this data in an optimal way, without losing critical information. The way that it is set up is to allow for "zooming in" on individual leukemias – we think that this is critical as the individual leukemic clones have different latencies and penetrance (hence it is difficult to pool mice for KM curve generation); an issue we also discuss. We have however in connection to our revision attempted to make a set of KM curves of these data (Supplementary Figure 1), which perhaps can aid clarity. We very much agree that "infinite" as used in Table 1 in our original manuscript is not ideal here, and have in our revised manuscript changed this to ">70 days" (which was the the time for final assessment).

Query: Immune-editing and immunoediting are used throughout. Consistency is recommended.

We thank the reviewer for this and have changed accordingly.

Query: Fig 2C: was this experiment done multiple times, and is this representative data, or pooled results, or a single experiment only?

The experiment demonstrating expansion of PD-1 expressing CD8+ cells paralleling leukemic expansion in secondary hosts was conducted once (using leukemic cells from clones II-16; one of the clones that we also used to assess the effect of CD8-depletion). While it would evidently be informative to have information on more clones, we feel the question of exact mechanism of rejection is not central in our work. Hence, we decided to do this only on one clone (and chose one that had been rather extensively characterized in our work; being initially transplanted to 11 WT mice, 7 Rag2 KO, 7 Rag2gc KO, and responding to CD8-depletion (5 treated and 5 non-treated mice).

Thank you for submitting your revised manuscript entitled "Immunoediting is Not a Primary Transformation Event in a Murine Model of MLL-ENL AML". As you will see, reviewer #2 thinks

that the revised version will be of value to the field, and we would be happy to publish your paper in Life Science Alliance pending final revisions necessary to meet our formatting guidelines.

REFEREE REPORTS

Reviewer #2 (Comments to the Authors (Required)):

The authors have improved the manuscript by including statistical analysis where requested, and have also added important recognition regarding some of the limitations of analysis. Although in depth sequencing analysis would have been interesting, the authors feel this is beyond the scope of the manuscript. The data are potentially important negative results and have relevance in the current clinical context where immunotherapy is being proposed for increasing subtypes of malignancy.